# Antifungal Activity of Antimicrobial Peptides and Proteins against *Aspergillus fumigatus*

**DOI:** 10.3390/jof6020065

**Published:** 2020-05-18

**Authors:** Eloise Ballard, Raif Yucel, Willem J. G. Melchers, Alistair J. P. Brown, Paul E. Verweij, Adilia Warris

**Affiliations:** 1Aberdeen Fungal Group, Institute of Medical Sciences, University of Aberdeen, Aberdeen AB25 2ZD, UK; eloiseballard@outlook.com; 2Iain Fraser Cytometry Centre (IFCC), Institute of Medical Sciences, University of Aberdeen, Aberdeen AB25 2ZD, UK; 3Cytomics Centre, Geoffrey Pope Building, University of Exeter, Exeter EX4 4QD, UK; r.yuecel@exeter.ac.uk; 4Centre for Expertise in Mycology and Department of Medical Microbiology, Radboud University Medical Centre, 6525 GA Nijmegen, The Netherlands; Willem.Melchers@radboudumc.nl (W.J.G.M.); paul.verweij@radboudumc.nl (P.E.V.); 5MRC Centre for Medical Mycology at the University of Exeter, Exeter 4EX 4QD, UK; a.j.p.brown@exeter.ac.uk

**Keywords:** *Aspergillus fumigatus*, antimicrobial peptides, histones, β-defensin-1, lysozyme

## Abstract

Antimicrobial peptides and proteins (AMPs) provide an important line of defence against invading microorganisms. However, the activity of AMPs against the human fungal pathogen *Aspergillus fumigatus* remains poorly understood. Therefore, the aim of this study was to characterise the anti-*Aspergillus* activity of specific human AMPs, and to determine whether *A. fumigatus* can possess resistance to specific AMPs, as a result of in-host adaptation. AMPs were tested against a wide range of clinical isolates of various origins (including cystic fibrosis patients, as well as patients with chronic and acute aspergillosis). We also tested a series of isogenic *A. fumigatus* isolates obtained from a single patient over a period of 2 years. A range of environmental isolates, obtained from soil in Scotland, was also included. Firstly, the activity of specific peptides was assessed against hyphae using a measure of fungal metabolic activity. Secondly, the activity of specific peptides was assessed against germinating conidia, using imaging flow cytometry as a measure of hyphal growth. We showed that lysozyme and histones inhibited hyphal metabolic activity in all the *A. fumigatus* isolates tested in a dose-dependent fashion. In addition, imaging flow cytometry revealed that histones, β-defensin-1 and lactoferrin inhibited the germination of *A. fumigatus* conidia.

## 1. Introduction

The human opportunistic fungal pathogen *Aspergillus fumigatus* is the main causative agent of pulmonary aspergillosis, which ranges from allergic syndromes and chronic infections to life-threatening invasive aspergillosis [1]. *A. fumigatus* conidia are the infective form of the fungus, which germinate to form hyphae, the invasive form of the fungus, to establish disease. *A. fumigatus* encounters various stresses in-host during infection or colonisation. In order to survive, the fungus must adapt to withstand these stresses [2,3]. These stresses include exogenous stresses, such as azole antifungals, and endogenous stresses that include oxidative stress, nutrient depletion and antimicrobial peptides and proteins (AMPs) [2,3].

AMPs are key components of the human defences against invading microorganisms. Over 100 human AMPs have been described, ranging in size from 10 to 149 amino acids, and have net charges ranging from −3 to +20 [4,5,6]. Many cationic AMPs are hypothesised to exert their antimicrobial activity via membrane interactions. One of the most prevalent models of interaction is the ‘carpet model’, whereby peptides adsorb parallel to the membrane and, once they reach a threshold concentration, they disintegrate the membrane in a detergent-like effect [7,8,9].

Previously, synthetic peptides derived from human lactoferrin, histatin and ubiquicidin have been shown to damage *A. fumigatus* hyphae in a dose-dependent manner [10]. In addition, drosomycin, a defensin-like molecule produced by *Drosophila melanogaster*, has been shown to possess broad spectrum activity against *Aspergillus* spp. [11]. Seemingly conflicting results have been reported for the cationic peptide LL-37, promoting as well as inhibiting the growth of *A. fumigatus* [12,13].

Development of resistance to AMPs has been observed in various bacteria, but has not been reported in fungi [13,14]. For example, in vivo serial passaging experiments have shown that *Staphylococcus aureus* can develop inducible resistance against lactoferrin B [14]. In addition, both clinical and reference strains of *S. aureus* have been shown to secrete LL-37-degrading proteinases [15,16,17].

In this study, the activity of specific AMPs against *A. fumigatus* was determined. As *A. fumigatus* primarily infects the lungs, AMPs with previously described abundance in the lungs were selected for investigation; these included β-defensin-1, LL-37, lactoferrin, lysozyme and histones [6,18,19,20,21,22,23,24,25,26,27,28,29,30]. Previous studies have also identified these specific AMPs to possess antifungal activity against specific fungi [12,13,31,32,33,34,35,36]. This information is summarised in Table 1.

This study aimed to characterise the anti-*Aspergillus* activity of these AMPs, and to determine whether clinical isolates obtained from various patient populations show differences in their susceptibilities to specific AMPs. In addition, we assessed whether changes in AMP susceptibility occur as a result of in-host adaptation. A wide range of clinical isolates were selected for testing, including our previously characterised isogenic strain series [37], environmental strains, strains from cystic fibrosis (CF) patients, and isolates from patients with chronic and acute invasive aspergillosis. Firstly, the metabolic activity of hyphae was assessed after short term incubation with AMPs. Secondly, the impact of AMPs on germinating conidia was assessed via imaging flow cytometry. Lysozyme and histones were shown to possess dose-dependent inhibitory activity against hyphae of clinical and environmental *A. fumigatus* isolates. In addition, imaging flow cytometry revealed that histones, β-defensin-1 and lactoferrin reduce hyphal growth in both environmental and clinical isolates. Hyphae and germinating conidia were selected for investigation, as these represent the invasive forms of the fungus.

## 2. Materials and Methods

### 2.1. A. Fumigatus Strains

Five *A. fumigatus* isolates recovered from CF patients, five from patients with chronic pulmonary aspergillosis (CPA) and five from patients with acute invasive aspergillosis (IA) were included in this study. These anonymised strains were obtained from the Centre of Expertise in Mycology at the Radboud University Medical Centre, the Netherlands. A series of 13 isogenic strains was also included, isolated from a single chronic granulomatous disease patient over a period of 2 years. We have previously characterised these isolates phenotypically and genotypically, revealing which have developed azole resistance and conidiation defects in-host throughout the course of infection [2,37,38,39]. We also included a range of *A. fumigatus* strains isolated from the soil in Scotland. The strains used in this study are summarised in Table 2. 

### 2.2. Preparation of Fungal Suspensions

*A. fumigatus* conidia from glycerol stocks frozen at −80 °C were spread onto Sabouraud dextrose agar in T75 culture flasks (Greiner Bio-One, Frickenhausen, Germany), and incubated at 37 °C. After 7 days, conidia were harvested via immersion in Phosphate Buffered Saline (PBS) containing 0.05% Tween-80 (Thermo Fisher Scientific, Renfrew, UK). Suspensions were passed through a sterile 40 µm strainer to remove hyphae and washed twice using PBS. Suspensions were counted using a Neubauer improved haemocytometer. Conidial suspensions were diluted in RPMI (1640 + Glutamax; Thermo Fisher Scientific).

### 2.3. Antimicrobial Peptides

The activity of specific AMPs was investigated; included were LL-37 (human; Bio-Techne, Newark, NJ, USA), lactoferrin (recombinant human version expressed in rice; Sigma Aldrich, Gillingham, UK), β-defensin-1 (recombinant human version; Abcam, Cambridge, UK), lysozyme (human; Sigma Aldrich) and histones (calf thymus; Roche, Welwyn Garden City, UK). All AMP solutions were prepared in PBS on the day of use. Initial test concentrations were prepared as follows: lysozyme 5–160 µM; histones 6.25–100 µg/mL; LL-37 5–50 µM; lactoferrin 10–40 µM and β-defensin-1 1–10 µM. Concentrations were selected based on reported physiological concentrations in bronchoalveolar lavage fluid and/or antifungal activity (see Table 1).

### 2.4. Determination of Metabolic Activity of Hyphae Using an XTT Assay

Flat bottomed 96 well plates (Thermo Fisher Scientific) were seeded with 5 × 10^4^ conidia in 50 µL RPMI and incubated for 16 h at 37 °C. Light microscopy was used to assess hyphal formation. Following hyphal formation, in order to remove potential proteases in the media, 96 well plates were centrifuged at 3000× *g*, and the media was removed and replaced with 50 µL RPMI. Subsequently, 50 µL of each individual AMP solution was added and incubated for 2 h at 37 °C. Initial test concentration ranges of AMPs were as follows: lysozyme 5–160 µM; histones 6.25–100 µg/mL; LL-37 5–50 µM; lactoferrin 10–40 µM and β-defensin-1 1–10 µM. In control wells, 50 µL PBS was added. After 2 h, a colorimetric XTT assay was used to assess fungal metabolic activity [40]. In brief, 100 µL 2× XTT solution ((200 µg/mL XTT (Sigma Aldrich) and 4.3 µg/mL Menadione (Sigma Aldrich)) was added to each well and incubated for 2 h at 37 °C. After incubation, plates were centrifuged, and 100 µl supernatant was transferred to a flat bottomed 96 well plate. Absorbance was measured at 450 nm using a VersaMax microplate reader (Molecular Devices, USA). The percentage of fungal metabolic activity in relation to control wells was subsequently calculated. Data were obtained from two independent experiments, each with triplicate readings.

Initial testing using a range of concentrations was performed with isolate V130-15 only, and subsequent investigation with specific concentrations of lysozyme and histones was performed with a larger collection of strains of different origins; all *Aspergillus* isolates listed in Table 2 were included for further investigation into lysozyme and histones. AMP combination experiments were performed using 6 representative strains (chronic granulomatous disease series: V130-15, V157-62; CF patient: 113-61; CPA: 151-06; IA: 120-76; environmental: ENV-S-22). See Table 2.

### 2.5. Determination of Germination Inhibition Using Imaging Flow Cytometry

Initial characterisation of fungal growth under control conditions using imaging flow cytometry was performed prior to investigating the impact of AMPs. U-bottomed 96 well plates (Thermo Fisher Scientific) were seeded with 1 × 10^5^ conidia per well in 50 µL RPMI and 50 µL of PBS was added to each well. Plates were incubated for 0, 8 and 10 h at 37 °C. After incubation, cells were suspended in 4% paraformaldehyde and incubated at room temperature for 15 min. Plates were subsequently centrifuged and the fixing solution was removed. Cells were resuspended in 50 µL PBS containing 0.2% Tween-20 (Thermo Fisher Scientific) to reduce clumping, transferred to 1.5 mL Eppendorf tubes and stored at 4 °C until analysis. Analysis was performed using the Amnis ImageStreamX MK II imaging flow cytometer (Luminex, Austin, TX, USA). Illumination source in the ImageStreamX MKII system is provided by a brightfield (BF) LED lamp, a 405 nm, 488 nm, 561 nm, and 642 nm excitation lasers with a separate 785 nm laser for side scatter light detection. The morphological features for circularity and length were determined by using the BF images of singe cells. Cellular parameters were measured using brightfield channels 1 and 9. Data analysis was performed using IDEAS software (version 6.2, Luminex, Austin, TX, USA). Gating strategies used to clean the data (removing SpeedBead™ controls and removing unfocused cells from the dataset) are shown in Appendix A.

Once the initial characterisation had been performed, the impact of AMPs on hyphal growth was assessed. The following AMP concentrations were selected: 80 µM lysozyme, 100 µg/mL histones, 10 µM β-defensin-1, 40 µM lactoferrin and 12.5 µM LL-37. These concentrations were determined based on our initial set of experiments in which anti-hyphal activity was assessed. U-bottomed 96 well plates were seeded with 1 × 10^5^ conidia per well in 50 µL RPMI. AMPs resolved in 50 µL PBS were added to specific wells. In control wells, only 50 µl PBS was added. Plates were incubated for 10 h at 37 °C. After co-incubation, cells were suspended in 4% paraformaldehyde and incubated at room temperature for 15 min. Plates were subsequently centrifuged and the fixing solution was removed. After fixing, cells were resuspended in 50 µL PBS containing 0.2% Tween-20, transferred to 1.5 mL Eppendorf tubes and stored at 4 °C until analysis. As in the initial characterisation, analysis was performed using the ImageStreamX MK II and data analysis was performed using IDEAS software as described in the Appendix A.

All imaging flow cytometry was performed at the Ian Fraser Cytometry Centre at the University of Aberdeen, UK. Initial characterisation experiments were performed in biological duplicates using isolate V130-15. Co-incubation with AMP was performed with 5 strains which were considered representative of the groups of strains used throughout the study (V130-15, 113-61, 151-06, 120-76 and ENV-S-22; see Table 2). These experiments were performed once for each strain.

### 2.6. Statistical Analysis

Statistical analysis was performed using GraphPad Prism (version 5). Statistical significance was assessed using a two-tailed *T* test. A *p* value < 0.05 was considered significant.

## 3. Results

### 3.1. Dose-Dependent Reductions in the Metabolic Activity of A. Fumigatus Hyphae by Lysozyme and Histones

Incubation with 5–160 µM lysozyme resulted in concentration-dependent decreases in the metabolic activity of *A. fumigatus* hyphae (see Figure 1A). Similarly, incubation with 6.25–100 µg/mL histones resulted in a concentration-dependent decrease in metabolic activity (see Figure 1B). Incubation with 5–50 µM LL-37, 10–40 µM lactoferrin or 1–10 µM β-defensin-1 did not result in reduced metabolic activity of hyphae in any isolates (including V130-15, 111-45, ENV-S-4, ENV-S-22 and ENV-S-12) tested. Based on these results, it was decided to proceed with analysis of lysozyme and histones at specific effective concentrations and a wider panel of *A. fumigatus* isolates.

### 3.2. Lysozyme Inhibits Hyphal Metabolic Activity in a Range of Clinical and Environmental Isolates

Lysozyme concentrations of 20 and 80 µM were selected for further investigation with a wider selection of *A. fumigatus* strains. As shown in Figure 2A, the series of 13 isogenic clinical isolates exhibited significantly reduced hyphal metabolic activity after incubation with 20 µM lysozyme (*p* < 0.0001). Mean metabolic activity of all 13 isolates after incubation with 20 µM lysozyme was 46% of the control. All isolates in the series showed further reduced metabolic activities after incubation with 80 µM lysozyme (*p* < 0.0001). Mean metabolic activity of the 13 isolates after incubation with 80 µM lysozyme was 11% compared to the control.

Within the isogenic series, isolates possessed differing susceptibilities to lysozyme. After incubation with 20 or 80 µM lysozyme, the range of metabolic activities was 31–60% and 5–15%, respectively. Specific isolates possessed significantly different susceptibilities to lysozyme. As an example, V130-15 showed to be more susceptible than V157-47, V157-60 or V157-62 to 20 µM lysozyme (*p* < 0.05). However, no correlations were observed between isolation date from the patient and lysozyme susceptibility. It is important to note that isolates in the isogenic series later isolated from the patient are shown further along the x axis on Figure 2A. The exact isolation dates are shown in Table 2.

As shown in Figure 2B, strains isolated from patients with cystic fibrosis, chronic infection, acute infection and environmental strains also showed reduced metabolic activity after incubation with 20 or 80 µM lysozyme (*p* < 0.0001). Metabolic activities after incubation with 20 or 80 µM lysozyme decreased to 31–65% and 5–25%, respectively. Interestingly, environmental isolates possessed on average lower metabolic activity after incubation with 20 µM lysozyme, when compared to either chronic, acute or CF strains (*p* < 0.0001).

### 3.3. Histones Inhibit Hyphal Metabolic Activity in a Range of Clinical and Environmental Isolates

Histone concentrations of 50 and 100 µg/mL were selected for further investigation with a wider selection of strains. The series of 13 isogenic clinical isolates possessed significantly reduced hyphal metabolic activities after incubation with 50 µg/mL histones (*p* < 0.0001) (Figure 3A). The mean metabolic activity of the strains after incubation with 50 µg/mL histones was 50% of the control metabolic activity. All isolates in the series showed further reductions in metabolic activity after incubation with 100 µg/mL histones (*p* < 0.0001). The mean metabolic activity of the strains after incubation with 100 µg/mL histones was 27% of the control metabolic activity. In our series of isogenic isolates (Figure 3A), there was no significant decrease in susceptibility to histones in strains isolated later from the patient.

In comparison to lysozyme, a greater variation in the susceptibilities of the isogenic isolates to histones was noted. After incubation with 100 µg/mL histone, isolates showed significantly different metabolic activities. As an example, V130-54 was significantly more susceptible to 100 µg/mL histone than isolates V130-15, V130-14, V157-39, V130-18, V157-47, V157-48, V157-62 and V157-61 (*p* < 0.01). Again, though, no trend towards decreased susceptibility was observed over time.

As shown in Figure 3B, strains isolated from patients with CF, CPA, IA and environmental strains also showed reduced metabolic activity after incubation with 50 or 100 µg/mL histone (*p* < 0.0001). The metabolic activities after incubation with 50 or 100 µg/mL histone were reduced to 35–66% and 6–45% compared to the untreated controls, respectively. No associations between the origin of the isolates and the susceptibility to histones could be made.

### 3.4. Additive Anti-Hyphal Effect of a Combination of Lysozyme and Histones

In order to better replicate the AMP stress encountered in the human lung, AMPs were also tested in combination. Six representative strains from each of the described groups were selected for these analyses.

The combination of 20 µM lysozyme plus 50 µg/mL histones showed increased antifungal activity against all of the isolates tested. The effects appear additive, but there was variability in the extent of this additive effect between the isolates. As shown in Figure 4, after incubation with either lysozyme or histones, ENV-S-22 showed reduced metabolic activities to 54% and 50%, compared to untreated controls, respectively. When exposing this strain to a combination of lysozyme plus histones, the metabolic activity was reduced to 38% compared to untreated controls (*p* = ns). The combination of AMPs appears to show minimal additive effects for this *A. fumigatus* strain.

On the other hand, isolate V130-15 had significantly less metabolic activity when incubated with the combination of lysozyme plus histones in comparison to lysozyme alone (*p* = 0.01). This combination of AMPs shows a greater additive effect for this strain. In addition, isolate 151-06 showed respective metabolic activities of 72% and 61% when incubated with lysozyme and histones alone, but 24% after incubation with them in combination. In comparison to incubation with histones alone, this strain showed significantly less metabolic activity than when incubated with the combination (*p* = 0.02). A similar trend was observed for isolate 113-61, which showed significantly lower metabolic activity after incubation with the AMP combination than with either histones or lysozyme individually (*p* = 0.02).

### 3.5. Characterisation of A. Fumigatus Cell Morphologies Using Imaging Flow Cytometry

Having investigated the activity of specific AMPs against hyphae, we decided to investigate the activity of these AMPs against germinating conidia. Analysis using imaging flow cytometry was selected as it enabled high throughput quantification between cell morphologies based on single cell images.

To achieve this, an initial characterisation of *A. fumigatus* cell morphologies, using imaging flow cytometry, was performed using a single isolate (V130-15). Incubation times of 0, 8 and 10 h were selected as representative time points for conidia, germinating conidia and hyphae [41]. Based on the time points and microscopic images produced from the ImageStreamX, gating analysis strategies for hyphae and conidia were developed (Appendix A). Morphological shape change is measured in the final graph as circularity and length features, presented in Figure 5A. Using this gating strategy, the percentage of the cells in the hyphae gate was quantified. This is shown in Figure 5B. At 0 h, 0% of the cells were hyphal, and this increased to on average 0.95% at 8 h. Finally, cells grown for 10 h had on average 13.7% of cells in the hyphae gate. Therefore, in comparison to either the 0 h or 8 h time point, the 10 h condition had significantly more cells in the hyphae gate (*p* < 0.0001).

### 3.6. Impact of Specific AMPs upon Fungal Growth

Having demonstrated that imaging flow cytometry can indeed be used to distinguish between conidial, germling and hyphal *A. fumigatus* cell morphologies, we proceeded to use this experimental set up to investigate whether incubation with specific AMPs influenced the germination and growth of five strains after 10 h. One strain from each of the patient groups was selected for investigation (from the chronic granulomatous disease series: V130-15; from the CF group: 113-61; from the CPA group: 151-06; from the IA group: 120-76 and from the environmental group: ENV-S-22).

As shown in Figure 6, there were significant differences in the percentage of hyphal cells after incubation with specific AMPs. Under control conditions, on average 41% of cells were in the hyphae gate. After incubation with histones, β-defensin-1, or lactoferrin, significantly fewer cells were observed in the hyphae gate; 3% (*p* < 0.0001), 27% (*p* = 0.04) and 21% (*p* = 0.003) respectively. No significant differences were observed after incubation with lysozyme or LL-37.

Mean hyphae lengths of the strains when grown under control conditions were 57.9 µM. When incubated with lysozyme, histones, β-defensin-1, lactoferrin, or LL-37, mean hyphal lengths were 53.6, 42.8, 51.9, 55.6, or 54.2 µM, respectively. It should be noted that this is the mean length of cells within the hyphae gate and does not consider the number of cells within this gate. Mean overall lengths of the strains when grown under control conditions was 33.4 µM. When incubated with lysozyme, histones, β-defensin-1, lactoferrin, or LL-37, mean overall lengths were 34.1, 12.2, 21.8, 22.8, or 30.7 µM, respectively. Mean lengths of hyphae and mean overall lengths are shown in Appendix A.

## 4. Discussion

In this study, a range of *A. fumigatus* strains of varying origin and azole resistance profiles were used to investigate the anti-Aspergillus activity of specific AMPs. We showed that lysozyme and histones both possess dose-dependent activity against *A. fumigatus* hyphae. In addition, imaging flow cytometry revealed that incubation with histones, β-defensin-1 or lactoferrin inhibited conidial germination.

Lysozyme is one of the most abundant AMPs in the human lungs [6,19,20,21]. Lysozyme is known to catalyse the hydrolysis of β-1,4 linkages between *N*-acetylmuramic acid and *N*-acetyl-d-glucosamine residues in peptidoglycan, and of glycosidic bonds in chitin [4,42,43]. Whether these enzymatic actions are responsible for the antimicrobial activity of lysozyme remains unclear. Lysozyme is also cationic in nature, and the associated positive charge may cause microbial membrane disruption [6,44]. In this study, we showed that lysozyme has a dose-dependent antifungal activity against *A. fumigatus* hyphae. Intriguingly, we did not observe any impact of lysozyme on germination or hyphal growth. It can be hypothesised that lysozyme specifically binds to a component only found on *A. fumigatus* hyphae. We predict that this target is absent or hidden on conidia or germinating (swollen) conidia, and as such lysozyme cannot bind or exert its antimicrobial activity on these morphologies.

Interestingly, this study showed that hyphae of the *Aspergillus* isolates in the isogenic clinical series displayed differing susceptibilities to lysozyme. Previously described differences in morphology and, most likely, differences in the cell wall composition of these isolates may account for these differences [37]. In addition, hyphae of the environmental *Aspergillus* isolates were shown to be more susceptible to lysozyme than the clinical isolates. Differences in the cell wall between these strains may confer these differences. The environmental isolates are unlikely to have been exposed to lysozyme previously, and this might account for their higher susceptibility to this AMP. This might indicate that *A. fumigatus* has the potential to undergo in-host adaptation to lysozyme. It is worth noting that, in theory, isolates from patients with acute invasive aspergillosis are likely to have spent a shorter time in-host compared to those from patients with chronic aspergillosis, and as such are expected to show comparable susceptibilities to lysozyme as shown for the environmental isolates. However, we were not able to show this in our study, which is most likely explained by the complexity of underlying mechanisms involved in in-host adaptation. Histones are evolutionary conserved proteins, which play essential roles in organisation and regulation of chromatin. In addition, histones are a key component of neutrophil extracellular traps (NETs) [45,46]. Although not traditionally thought of as AMPs, histones have been shown to possess antimicrobial activity against bacteria, fungi and viruses [34,45,47,48]. In *Escherichia coli*, the antimicrobial activity of histones has been reported to be due to membrane disruption [49]. It is unclear whether histones also disrupt the membranes of fungi. Excitingly, our results show that histones possess anti-*Aspergillus* activity against germinating conidia, as well as anti-hyphal activity. The mechanism via which this occurs is unknown at present. Our study shows that hyphae of isolates in the isogenic clinical series display differing susceptibility to histones. Known differences in the morphology of these isolates may confer these differences [37]. The environmental isolates were not shown to be more susceptible to histones in comparison to the clinical isolates tested (this trend was observed with lysozyme). As lysozyme and histones likely do not possess the same mechanism of action, differences in the hyphal structure between the *Aspergillus* isolates may explain this.

Previously, β-defensin-1 has been reported to inhibit the growth of *C. albicans* [36]. In this study, imaging flow cytometry revealed that β-defensin-1 can significantly restrict fungal growth, but, on the other hand, is not active against already-formed *A. fumigatus* hyphae. It can be hypothesised that the target of this peptide is present on germinating conidia, but not hyphae. The expression of β-defensin-1 by human bronchial epithelial cells has been shown to be induced upon exposure to *A. fumigatus* [50,51]. It is conceivable that, in response to the inhalation of conidia (the infective particle of *A. fumigatus*), increased expression of β-defensin-1 is triggered, which in turn inhibits the invasive hyphal growth of the fungus.

Lactoferrin is an iron-sequestering cationic protein with reported antimicrobial activity against viruses, bacteria and fungi [18]. This activity has been reported to be both iron-dependent and independent [18,22]. Xu et al. reported that, after exposure to lactoferrin, *C. albicans* exhibits profound cell wall changes, including swelling and collapse [52]. We showed that lactoferrin has inhibitory activity against germinating conidia, but not hyphae. It can be hypothesised that differences in the cell wall between germinating conidia and hyphae result in differential antifungal activities of lactoferrin. Additionally, germination might be more dependent on iron compared to hyphal growth. Previously, Lupetti et al. showed that synthetic fragments of lactoferrin possess activity against *A. fumigatus* hyphae [10]. Differences in structure between the synthetic fragments used previously and the full peptide utilised in this study may explain these apparently contrasting results.

LL-37 did not show any antifungal effect (at 5–50 µM) in either of our two experimental designs used in this study. These experimental designs used differing morphologies at the start of the experiment (e.g., hyphae and conidia) and various incubation periods (2 and 10 h), as well as differing read-outs (XTT assay assessing metabolic activity, and imaging flow cytometry assessing germination). Recently, 1 µM LL-37 has been reported to promote the growth of *A. fumigatus* when grown in Sabouraud dextrose broth for 24 and 48 h [12]. In contrast with this, another recent study reported that 1–20 µM LL-37 inhibits mycelial growth of *A. fumigatus* in a dose-dependent manner when grown in RPMI for 12 h [13]. These contrasting observations are not easily explained, although differences between strains, AMP concentrations and experimental conditions probably account for these contrasting results. Our data are consistent with the latter study, suggesting that LL-37 does not significantly affect the development of *A. fumigatus* hyphae.

In vivo concentrations of specific AMPs have been reported to vary due to an individual’s age and/or their underlying conditions. For example, CF patients have been reported to have higher levels of lysozyme and lactoferrin in bronchoalveolar lavage fluid in comparison to healthy controls [53,54,55]. The bronchoalveolar lavage fluid of new born infants during pulmonary or systemic infection has been described to have significantly increased levels of β-defensin-1, β-defensin-2 and LL-37 in comparison to healthy controls [56]. Therefore, it can be hypothesised that clinical isolates originating from different patient populations may have been exposed to different levels or combinations of AMPs in-host. This could perhaps drive the emergence of different resistance profiles to AMPs in these different isolates. We did not observe such a trend towards AMP resistance in our unique series of isogenic isolates, despite the fact that we have shown that these isolates underwent other phenotypic and genotypic changes [37]. This may be because they received minimal exposure to these AMPs in-host, or that AMP resistance is unlikely to develop. If this holds true, these findings support the development of AMP-based antifungal therapy, as our data indicate that the acquisition of resistance is either not likely or a much slower process than the acquisition of antifungal drug resistance. In vitro experimental evolution experiments, during which the fungus is exposed to AMPs over a longer period, could provide useful insights into this.

Finally, our results showed that the azole resistance profile of the *A. fumigatus* isolates did not impact their susceptibility to AMPs. This is not surprising, as the targets of AMPs are distinct to the targets of azole antifungal agents. Although not unexpected, this trend is still important to note, as it indicates that peptide-based antifungals may have potential to be used for infections caused by azole-resistant *A. fumigatus* [57,58].

## 5. Conclusions

In conclusion, both the AMPs’ lysozyme and histones show dose-dependent activities against *A. fumigatus* hyphae, irrespective of the isolate’s azole resistance profile. In addition, histones, lactoferrin and β-defensin-1 display inhibitory activities against germinating conidia. This study showed that the environmental isolates tested are more susceptible to lysozyme than the clinical isolates tested. However, there was not any clear evidence for in-host acquisition of resistance to these specific AMPs. This absence of development of resistance to AMPs could make AMP-based therapy a viable antifungal option.

## Figures and Tables

**Figure 1 jof-06-00065-f001:**
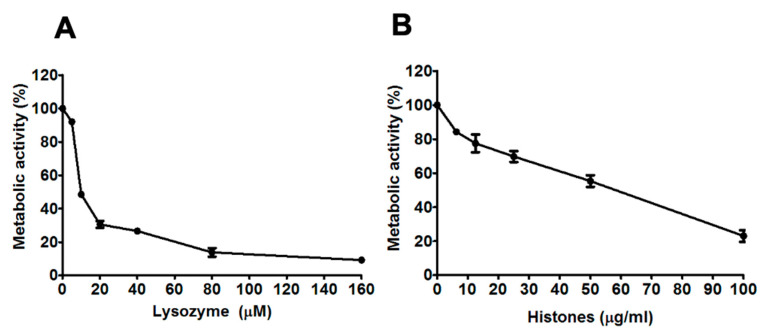
Concentration-dependent antifungal activity of lysozyme and histones against *A. fumigatus* V130-15 (**A**) Metabolic activity after 2 h incubation with lysozyme (5–160 µM). (**B**) Metabolic activity after 2 h incubation with histones (6.25–100 µg/mL). In each experiment, hyphae were grown overnight in RPMI and the media was refreshed prior to adding the lysozyme or histones in phosphate buffered saline. Metabolic activity was determined by XTT assay. Data was obtained in two independent experiments, each with triplicate readings; mean values ± SD are shown.

**Figure 2 jof-06-00065-f002:**
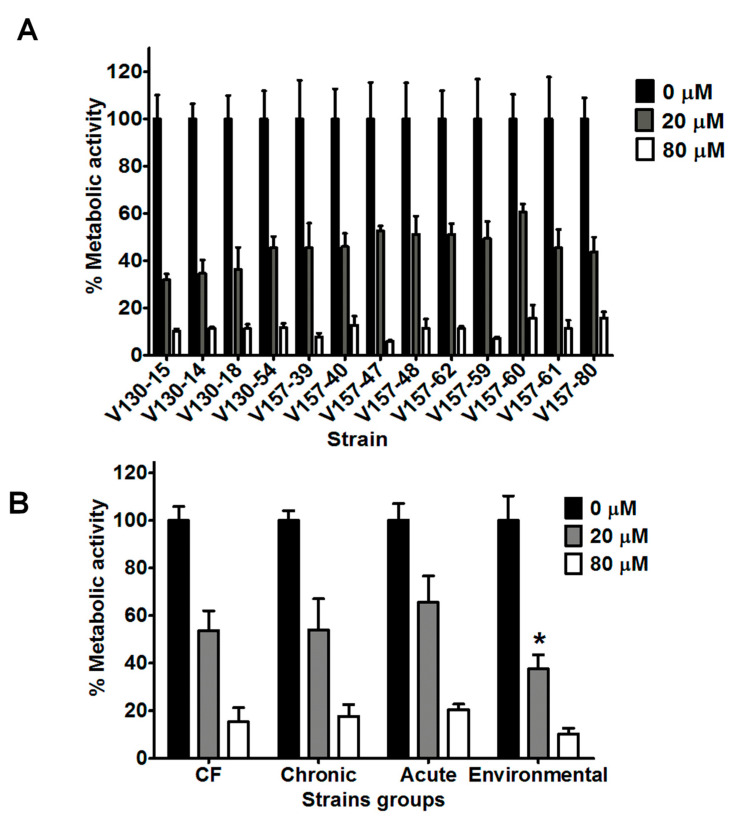
Antifungal activity of lysozyme against a range of *A. fumigatus* isolates. (**A**) Metabolic activity of the series of clinical isolates after 2 h incubation with lysozyme (20 or 80 µM). (**B**) Metabolic activity of strains of various clinical and environmental origins (data from 5 strains is combined into each group) after 2 h incubation with lysozyme (20 or 80 µM). In each experiment, hyphae were grown overnight in RPMI and the media was refreshed prior to adding the peptide in phosphate buffered saline. Metabolic activity was determined by XTT assay. Data were obtained in two independent experiments, each with triplicate readings; mean values ± SD are shown. * denotes statistical difference when compared to CF, chronic and acute means for incubation with 20 µM lysozyme (two tailed *t*-test; *p* < 0.0001).

**Figure 3 jof-06-00065-f003:**
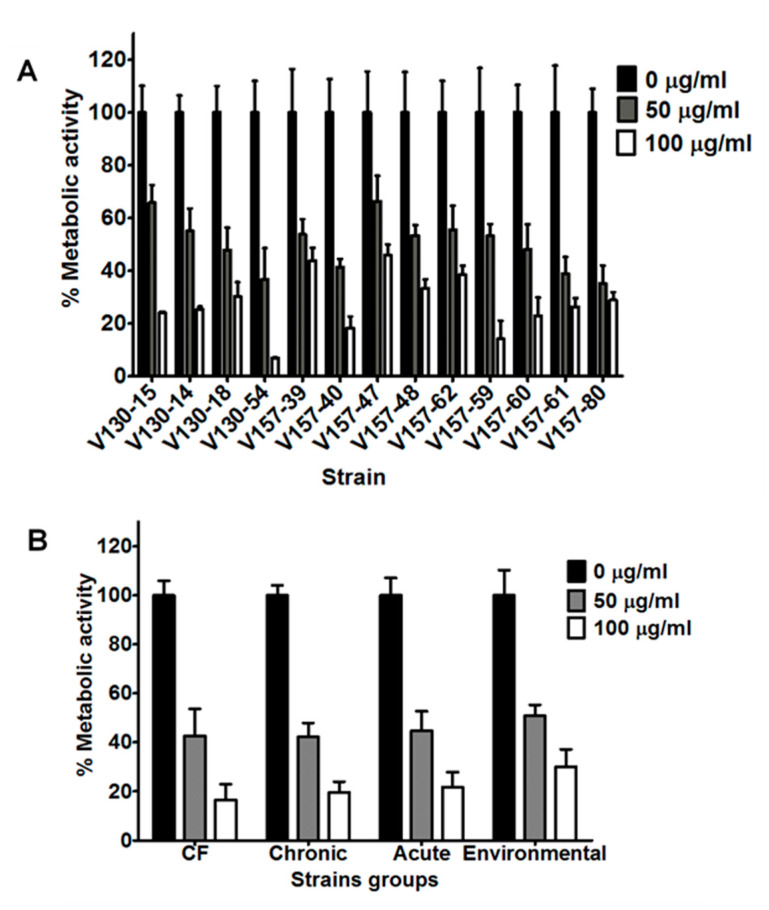
Antifungal activity of histones against a range of *A. fumigatus* isolates. (**A**) Metabolic activity of a series of isogenic isolates after 2 h incubation with histones (50 or 100 µg/mL). (**B**) Metabolic activity of strains of various clinical and environmental origins (data from 5 strains is combined into each group) after 2 h incubation with histones (50 or 100 µg/mL). Hyphae were grown overnight in RPMI and the media was refreshed prior to adding the peptide in phosphate buffered saline. Metabolic activity was determined by XTT assay. Data were obtained in two independent experiments, each with triplicate readings; mean values ± SD are shown.

**Figure 4 jof-06-00065-f004:**
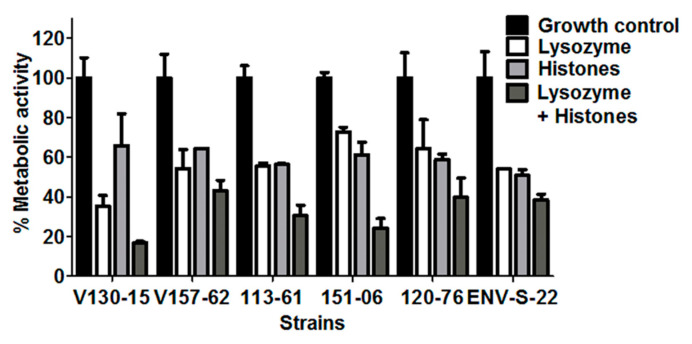
Antifungal activity of lysozyme, histones, and lysozyme plus histones combined against six *A. fumigatus* isolates. Metabolic activity of the isolates after 2 h incubation with lysozyme (20 µM), histones (50 µg/mL) or in combination is shown. In each experiment, hyphae were grown overnight in RPMI and the media was refreshed prior to adding the peptide in phosphate buffered saline. Metabolic activity was determined by XTT assay. Data was obtained in two independent experiments, each with triplicate readings; mean values ± SD are shown.

**Figure 5 jof-06-00065-f005:**
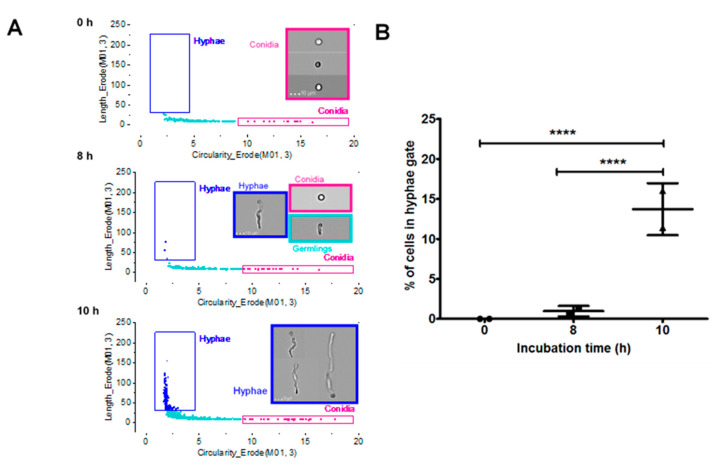
Initial characterisation of fungal morphologies of *A. fumigatus* (V130-15) using imaging flow cytometry. (**A**) Discrimination between fungal morphologies using length and circularity features in IDEAS analysis software: conidia, pink box; germlings, cyan box; hyphae, dark blue box. (**B**) The percentage of cells in the hyphae gate shown at the different timepoints. Conidia suspended in RPMI were incubated for 0, 8 or 10 h at 37 °C. Cells were fixed in 4% paraformaldehyde and analysed using imaging flow cytometry. Data was obtained in duplicate, each with triplicate readings; mean values ± SD are shown. **** *p* < 0.0001 as assessed by two-tailed *T* test.

**Figure 6 jof-06-00065-f006:**
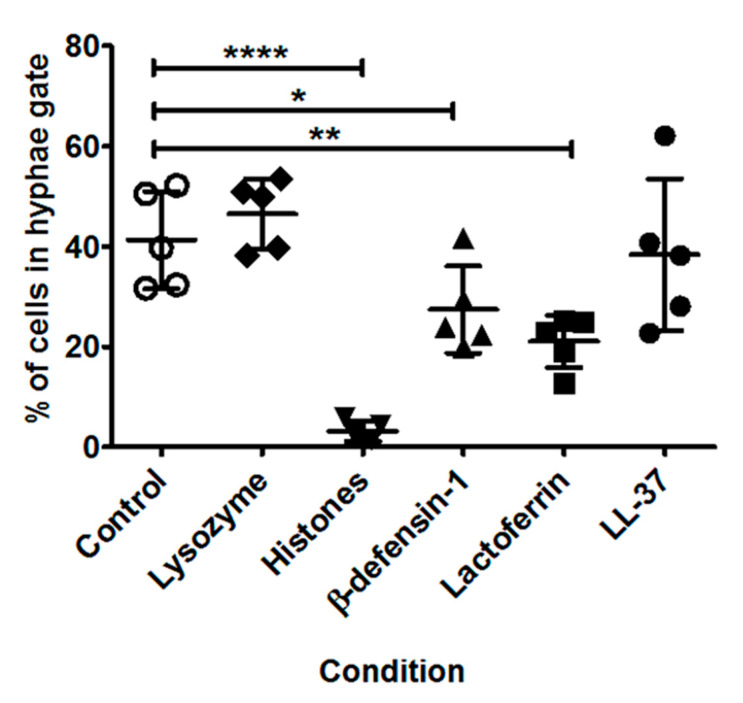
Average percentage of cells in the hyphae gate for each AMP for the five *A. fumigatus* strains analysed combined. Strains were incubated with antimicrobial peptides and proteins (AMPs) for 10 h at 37 °C. AMP concentrations were: 80 µM lysozyme, 100 µg/mL histones, 10 µM β-defensin-1, 40 µM lactoferrin and 12.5 µM LL-37. After incubation, cells were fixed in 4% paraformaldehyde and analysed using imaging flow cytometry. The data represents a single experiment per strain and mean values for all strains ± SD are shown. * *p* < 0.05; ** *p* < 0.005 and **** *p* < 0.0001 as assessed by two-tailed T test.

**Table 1 jof-06-00065-t001:** Summary of physiological concentrations in vivo and reported antifungal activity of the antimicrobial peptides and proteins selected for investigation.

Antimicrobial Peptide or Protein	Reported Physiological Concentrations in Bronchoalveolar Lavage Fluid	Reported Antifungal Activity
Lysozyme	5–70 µM 1 [19,21]	Inhibitory activity against *Histoplasma capsulatum* reported at 0.06 µM [31,32], and against *Aspergillus* and *Candida* spp. at 5 µM [33]
Histones	5–150 µg/mL [29,30]	Inhibitory activity against *Candida albicans* reported at 26 µg/mL [34]
LL-37	0.5–10 nM [25,26,27,28]	Inhibitory activity against *A. fumigatus* at 20 µM [15] and growth promotion activity against *A. fumigatus* at 1 µM [12,13]
lactoferrin	1–10 µM [18,19,20,21,22]	Inhibitory activity against *C. albicans* at 0.25–2.5 µM [35]
β-defensin-1	0.2–2 nM [6,23]	Inhibitory activity against *C. albicans* at 7 µM [36]

**Table 2 jof-06-00065-t002:** Origins and minimum inhibitory concentrations (MIC) of the *A. fumigatus* isolates used in this study.

Strain	Origin	MIC (mg/L)			*cyp51A* Genotype
		Itraconazole	Voriconazole	Posaconazole	
V130-15 ^a^	Strain series from a single chronic granulomatous disease patient	1	1	0.25	Wild type
V130-14 ^b^		1	1	0.25	Wild type
V130-18 ^b^		4	4	0.5	Wild type
V130-54 ^b^		>16	1	0.125	Wild type
V157-39 ^c^		>16	1	>16	G54R
V157-40 ^c^		>16	1	>16	G54V
V157-47 ^c^		>16	2	>16	P216L
V157-48 ^c^		>16	2	>16	P216L
V157-62 ^c^		>16	8	>16	M220R
V157-59 ^d^		>16	4	>16	M220R
V157-60 ^d^		>16	4	>16	M220R
V157-61 ^d^		>16	4	>16	M220R
V157-80 ^e^		>16	1	>16	P216L
107-49	Cystic fibrosis patients	2	0.25	0.5	N.D.
113-61		1	1	0.25	N.D.
059-23		2	0.5	0.5	N.D.
102-25		1	0.25	0.5	N.D.
104-10		2	0.125	0.5	N.D.
102-14	Chronic infections	1	0.25	0.5	N.D.
151-06		>16	8	1.0	TR_34_/L98H
219-21		0.5	1	0.125	N.D.
226-53		>16	2	1	TR_34_/L98H
094-75		1	1	0.5	N.D.
111-46	Acute infections	1	4	0.25	N.D.
111-60		0.5	0.5	0.063	N.D.
115-45		0.5	1	0.125	N.D.
116-50		2	2	0.25	N.D.
120-76		0.25	0.5	0.063	N.D.
ENV-S-4	Environment (isolated from soil in Scotland, UK)	0.5	0.25	0.006	N.D.
ENV-S-22		>16	4	0.125	TR_34_/L98H
ENV-S-6		>16	4	1	TR_34_/L98H
ENV-S-12		>16	4	1	TR_34_/L98H
ENV-S-20		>16	4	0.5	TR_34_/L98H

Dates of isolation: ^a^, 22/11/2011; ^b^, 25/11/2011; ^c^, 09/12/2013; ^d^, 12/12/2013; ^e^, 19/12/2013. N.D.: *cyp51A* genotype not determined because of phenotypic profile.

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
