# Peer review of "Antifungal Activity of Antimicrobial Peptides and Proteins against Aspergillus fumigatus"

_jof, 2020, doi:10.3390/jof6020065_

Round 1
Reviewer 1 Report
In reviewer opinion introduction is insufficient. It will be beneficial to describe aspergillosis, its statistics – how common is it; what is mortality rate; what is a risk group? (a few sentences to make a context/background; but no too lengthy) Moreover, authors should underline importance of hyphae (peptides anti-hyphal effect is tested and fungal morphology). Is it a virulence factor? Lactoferrin is not a peptide but glycoprotein (80 kDa), again Histones are formally proteins; lysozyme also can be defined as a protein. Please, be more specific and verify your article to name compounds properly. In the view of fact that some molecules used in this study are peptides but some of them are proteins title “Antifungal activity of antimicrobial peptides against Aspergillus fumigatus” should be changed to “Antifungal activity of antimicrobial peptides and proteins against Aspergillus fumigatus”.
Line 27: „human β-defensin 1” instead „β-defensin-1”; please introduce its full name and abbreviation “HBD-1”
Line 37: author (Human Antimicrobial Peptides and Proteins Guangshun Wang) stated that net charge is between −3 and +20
Line 43: It is not clear why authors mention drosomycin. Please, focus on peptides used in this study and introduce reader to this issue. Selected peptides have specific modes of action and it should be clarified in introduction to motivate your choice.
Line 49: Peptide name is “lactoferricin B” not “lactoferrin”; please, give some example of lactoferrin antifungal (A. fumigatus) activity in introduction. Authors can also underline the difference between lactoferrin and lactoferricin B.
Table 1: Try once again to find information on antifungal activity of selected peptides against A. fumigatus – i.e. Charmaine M. Woods, David N. Hooper, H. Ooi, Lor-Wai Tan, A. Simon Carney, Human lysozyme has fungicidal activity against nasal fungi.
Line 50, 73, 80, 159: names of bacterial/fungal species – italic
Line 76: “Radboud University Medical Centre, in Netherlands”
Lines 99-100, and Figure 1: Is it possible to express concentration (perform calculations to unify units) in two units simultaneously - µg/ml and µM? Only Histones are in µg/ml.
Lines 103-104: Could you please use symbols “×” and “°” instead “x” and “o”?
Lines 104 and 106/107,235: Check if volume/concentration is correct (µl).
Line 110: reference 38 should be in square brackets
Line 133: Is it possible to give more specific information about laser used?
Line 146: “data” instead “ata”
Figure 2 and 3: Please, use introduced abbreviations: “CPA” instead “chronic” and “IA” instead “Acute”.
Line 302, 304; figures S2,S3: Verify hyphal length units (nm?)
Author Response
In reviewer opinion introduction is insufficient. It will be beneficial to describe aspergillosis, its statistics – how common is it; what is mortality rate; what is a risk group? (a few sentences to make a context/background; but no too lengthy) Moreover, authors should underline importance of hyphae (peptides anti-hyphal effect is tested and fungal morphology). Is it a virulence factor? Lactoferrin is not a peptide but glycoprotein (80 kDa), again Histones are formally proteins; lysozyme also can be defined as a protein. Please, be more specific and verify your article to name compounds properly. In the view of fact that some molecules used in this study are peptides but some of them are proteins title “Antifungal activity of antimicrobial peptides against Aspergillus fumigatus” should be changed to “Antifungal activity of antimicrobial peptides and proteins against Aspergillus fumigatus”.
> We have added an additional sentence in the introduction to provide a brief background on the disease pulmonary aspergillosis (line 32-34). Additional information on the importance of the hyphal form of the fungus has also been added (line 35-36) as well as clarification as to why hyphae and germinating conidia were selected for investigation (line 75-76). Title has been changed as suggested.
Line 27: „human β-defensin 1” instead „β-defensin-1”; please introduce its full name and abbreviation “HBD-1”
> In section 2.3 we have provided the details of the AMPs used to provide the readers with the origin of the AMPs; as can be noticed in this paragraph, LL-37 and lysozyme are from human origin, the lactoferrin and β-defensin-1 are recombinant human versions, and the histones originate from calf thymus. We have chosen not to repeat this each time we are referring to specific AMPs.
Line 37: author (Human Antimicrobial Peptides and Proteins Guangshun Wang) stated that net charge is between −3 and +20
> we have to apologize for this mistake and have corrected this accordingly in line 42:
Old: ranging from -3 to +16
Revised: ranging from -3 to +20 (line 44)
Line 43: It is not clear why authors mention drosomycin. Please, focus on peptides used in this study and introduce reader to this issue. Selected peptides have specific modes of action and it should be clarified in introduction to motivate your choice.
> The reason to refer to this study is to provide the background information with respect to what is known about AMPs with activity against Aspergillus spp. Drosomycin is a defensin-like molecule (like β-defensin-1), and we have therefore decided to include this study in the introduction.
Line 49: Peptide name is “lactoferricin B” not “lactoferrin”; please, give some example of lactoferrin antifungal (A. fumigatus) activity in introduction. Authors can also underline the difference between lactoferrin and lactoferricin B.
> As we have been focussing on lactoferrin, and not lactoferricin B, we don’t see a good reason to detail the difference between these two. We do thank the reviewer with respect to his comments on the fact that we used not only antimicrobial peptides in our studies, but also proteins, as lactoferrin is a protein and not a peptide. We have amended the title accordingly to recognize this, as well as in line 39.
> We have not been able to find a study in which the antifungal effect of lactoferrin as such has been assessed against A. fumigatus. We do acknowledge the paper in which peptides derived from lactoferrin have been assessed against A. fumigatus (see line 46-47 and reference 10). Table 1 is summarizing the published work with respect to antifungal activity of the AMPs as used in our study.
Table 1: Try once again to find information on antifungal activity of selected peptides against A. fumigatus – i.e. Charmaine M. Woods, David N. Hooper, H. Ooi, Lor-Wai Tan, A. Simon Carney, Human lysozyme has fungicidal activity against nasal fungi.
> we thank the reviewer for providing us with this paper, in which lysozyme showed to have antifungal activity against Aspergillus and Candida species. We have included this information in Table 1 and have added the reference.
Line 50, 73, 80, 159: names of bacterial/fungal species – italic
> we apologize for this incorrectness and have reviewed the manuscript in detail to ensure all names of bacterial/fungal species are written in ‘italic’ font.
Line 76: “Radboud University Medical Centre, in Netherlands”
> thanks for noticing the omission ‘the Netherlands’, this has been added in line 82 of the revised manuscript.
Lines 99-100, and Figure 1: Is it possible to express concentration (perform calculations to unify units) in two units simultaneously - µg/ml and µM? Only Histones are in µg/ml.
> We do agree with the reviewer, it would be nice to have unified units, but this decision was made as the histones used are a mixture of histone subunits. In this case using µg/ml is more accurate as molarity cannot be accurately calculated. Therefore, we propose to leave this as is.
Lines 103-104: Could you please use symbols “×” and “°” instead “x” and “o”?
> we have corrected the use of superscript (‘o’) throughout the manuscript, which was lost during formatting the manuscript in the required template.
Lines 104 and 106/107,235: Check if volume/concentration is correct (µl).
> the use of our symbol ‘µ’ was lost somehow in the formatting of the manuscript in the required template, but have corrected this in the revised version. It should read as ‘µl’.
Line 110: reference 38 should be in square brackets
> thanks for noticing this error, we have used the square brackets to indicate we are referring to reference 38 (which is now number 40 due to the addition of 2 references).
Line 133: Is it possible to give more specific information about laser used?
> Sure, and we have added the following information to the manuscript: ‘Illumination source in the ImageStreamX MKII system is provided by a brightfield (BF) LED lamp, a 405nm, 488 nm, 561nm, 642nm lasers excitation lasers with separate 785nm laser for side scatter light detection. The morphological features for circularity and length were determined by using the BF images of singe cells.’ (line 138-141)
Line 146: “data” instead “ata”
> we have added the missing ‘d’ in ‘data’
Figure 2 and 3: Please, use introduced abbreviations: “CPA” instead “chronic” and “IA” instead “Acute”.
> we would like to prefer to the terminology as used to provide an easy to read format, without the need to search for the meaning of the abbreviations
Line 302, 304; figures S2,S3: Verify hyphal length units (nm?)
> As indicated in the figure axis, the units of hyphal length are µM (e.g. MicroMeter)
Reviewer 2 Report
The manuscript entitled "Antifungal activity of antimicrobial peptides against Aspergillus fumigatus" by Ballard et al., describes the activity of lysozyme, histones, LL-37, lactoferrin and β-defensin-1, against A. fumigatus, by evaluation of metabolic activity of hyphae and conidial germination inhibition.
Minor comments and questions:
Revise the italic for genera and species of microorganisms; in vivo; cyp 51A
Check the space: 2h or 2 h (example)
Line 92: “………..diluted as required in RPMI…..”. Required by Who?
If it is not a deformation in the presentation, please change:
Line 103 and 126: 104 instead 104 and 105 instead 105, respectively. Revise all.
Line 107 and others: 37ºC instead 37oC. Revise all.
Line 110: [38] instead 38
Line 211: ”…………….histones in strains isolated…..”
Line 172: “3.2. Lysozyme inhibits ……………..clinical and environmental isolates” like in Histones
Table 1:
In LL-37: the authors have …..against A. fumigatus at 20 µM [14], but the reference is about S. aureus, is it correct?
Table 2:
Suggestion: “Origin………….concentrations (MIC) of the……..study”
Figure 1A:
Metabolic instead Meabolic
Figure1: Suggestion: “…………….against A. fumigatus V130-15. (A) Metabolic activity after 2 h …………(B) Metabolic activity after 2 h……
Figure2: Suggestion: “…………….(A) Metabolic……series of isogenic isolates……….. adding the peptide in PBS…….” Like in figure 3 and 4
Figure3: Suggestion: “…………….(B) Metabolic……various clinical and environmental…….” Like in figure 2
Figure4: Suggestion: “…………….lysozyme plus histones…….”
Figure5: Suggestion: Add the number of the isolate used
Author Response
Revise the italic for genera and species of microorganisms; in vivo; cyp 51A
> we apologize for this incorrectness and have reviewed the manuscript in detail to ensure all names of bacterial/fungal species are written in ‘italic’ font, as well ‘in vivo’ and ‘cyp51A’.
Check the space: 2h or 2 h (example)
> we have reviewed the whole manuscript and kept the use of the space uniformly throughout the manuscript.
Line 92: “………..diluted as required in RPMI…..”. Required by Who?
> thanks for drawing our attention to this, and we have deleted ‘as required’ in this sentence as it doesn’t refer to a specific procedure as such.
If it is not a deformation in the presentation, please change:
Line 103 and 126: 104 instead 104 and 105 instead 105, respectively. Revise all.
> we have corrected the use of superscript throughout the manuscript, which was lost during formatting the manuscript in the required template.
Line 107 and others: 37ºC instead 37oC. Revise all.
> we have corrected the use of superscript (‘o’) throughout the manuscript, which was lost during formatting the manuscript in the required template.
Line 110: [38] instead 38
> thanks for noticing this error, we have used the square brackets to indicate we are referring to reference 38 (number 40 in the revised version).
Line 211: ”…………….histones in strains isolated…..”
> we have updated this slightly confusing wording
Line 172: “3.2. Lysozyme inhibits ……………..clinical and environmental isolates” like in Histones
> we have updated this wording, and added the word ‘environmental’
Table 1:
In LL-37: the authors have …..against A. fumigatus at 20 µM [14], but the reference is about S. aureus, is it correct?
> this is indeed a mistake, reference 12 is the correct one (and not 14) in which this data is retrieved from, and has been corrected in the revised manuscript (please note in the revised manuscript ref 12 has become ref 13).
Table 2:
Suggestion: “Origin………….concentrations (MIC) of the……..study”
> thanks for this suggestion, and we have added the abbreviation ‘(MIC)’ to the title of the table in line 88
Figure 1A:
Metabolic instead Meabolic
> apologies for this and we have corrected this typo in the revised manuscript
Figure1: Suggestion: “…………….against A. fumigatus V130-15. (A) Metabolic activity after 2 h …………(B) Metabolic activity after 2 h……
> we have taken over this suggestion and amended accordingly in the figure legend
Figure2: Suggestion: “…………….(A) Metabolic……series of isogenic isolates……….. adding the peptide in PBS…….” Like in figure 3 and 4
> we have changed ‘AMP’ to ‘peptide’ as suggested
Figure3: Suggestion: “…………….(B) Metabolic……various clinical and environmental…….” Like in figure 2
> we have added ‘environmental’ as suggested
Figure4: Suggestion: “…………….lysozyme plus histones…….”
> we have changed ‘and’ to ‘plus’ as suggested
Figure5: Suggestion: Add the number of the isolate used
> The number of the used isolate has been added in the figure legend as well as in the text (line 274)
Reviewer 3 Report
The authors present an interesting study on the anti-Aspergillus fumigatus activity of selected antimicrobial peptides (AMP). The topic is of special interest, because, despite the avaliabilty of potent anti-fungal agents, invasive fungal infections are still an important cause of morbidity and mortality in immunocompromised patients such as hematopoietic stem cell transplant recipients. Therefore, new therapeutic appriaches are needed.
Although the study is well designed and the results are clearly presented, I would like to address some questions and comments to the authors.
- The authors testest the Aspergillus fumigatus strain V130-15 only. Based on these preliminary results they decided to exclude LL-37, lactoferrin and β-defensin from further experiments. However, as the authors correctly state that one has to „determine whether clinical isolates obtained from various patient populations show differences in their susceptibilities to specific AMPs“ (line 59 ff) it would be important to test LL-37, lactoferrin and β-defensin with at least one addition Aspergillus strain. These experiments would decrease the risk of missing important positive results. Maybe the authors have already some data they could present at least in the supplementary data section.
- In my opinion it is problematic to pool data from different strains. Although the presented SD indicate no major differences, a more differentiated presentation of the results would be beneficial (e.g. supplementary data).
- In discussion the authors state that „A. fumigatus hast he potential to undergo in-host adaptation to lysozymes“ (line 334). On the the other hand, they state that they „did not observe (such) a trend towards AMP resistance“ in their experimental setting (line 385 ff). This is confusing for the reader and should be clarified togther with experimental data.
- The authors state that the „environmental isolates possessed on average lower metabolic activity after incubation with 20 μM lysozyme when compared to either chronic, acute or CF strains“ (line 200) and explain that as follows: „The environmental isolates are unlikely to have been exposed to lysozyme previously and this might account for their higher susceptibility to this AMP. This might indicate that A. fumigatus has the potential to undergo in-host adaptation to lysozyme. (line 332 ff). If so, should then A. fumigatus strains isolated from the patients with acute invasive aspergillosis show similar susceptibility pattern as the evironmental strains since it is most likely that the origin of these infections are enviromental strains and not „clinical“ strains? This should be clarified in more detail.
- Why does the percentage of hyphae differ between the initial experiments (15 %, fig 5) and further tests (40%, Fig 6) ?
- The authors refer to data from other groups showing that CF patients have higher levels of lysozyme and lactoferrin in bronchoalveolar lavage fluid in comparison to healthy controls. In addition, increased levels of β-defensin-1, β-defensin-2 and LL-37 have been described in newborns during pulmonary or systemic infection in comparison to healthy controls. Is there any data on the AMP levels in immunocompromised patients such as hematopoietic stem cell transplant recipients? What does that mean for the therapeutic use of AMPs against fungal infections? Which patient population could benefit from AMP anti-fungal therapy?
- Can the authors provide some references showing a correlation of AMP levels and severity/cource of infections?
Author Response
The authors testest the Aspergillus fumigatus strain V130-15 only. Based on these preliminary results they decided to exclude LL-37, lactoferrin and β-defensin from further experiments. However, as the authors correctly state that one has to „determine whether clinical isolates obtained from various patient populations show differences in their susceptibilities to specific AMPs“ (line 59 ff) it would be important to test LL-37, lactoferrin and β-defensin with at least one addition Aspergillus strain. These experiments would decrease the risk of missing important positive results. Maybe the authors have already some data they could present at least in the supplementary data section.
> We started our studies by using the clinical isolate V130-15 to determine an optimal concentration of AMPs to be used in follow-up experiments as described in our manuscript. To confirm that L-L37, lactoferrin and β-defensin-1 did not have any antifungal activity against A. fumigatus, we tested an additional 3 environmental isolates (ENV-S-4, ENV-S-22, ENV-S-12), 1 additional clinical isolates (111-45) and 1 lab strain (AF293). We have clarified this in the text (see line 171-172). We don’t feel it adds value to include an additional graph to the supplementary data showing these negative results.
In my opinion it is problematic to pool data from different strains. Although the presented SD indicate no major differences, a more differentiated presentation of the results would be beneficial (e.g. supplementary data).
> We would be happy to provide the results obtained with the individual data, but the problem we are facing at the moment is that although we have most data electronic, we need to go back to our lab books to ensure we can provide a validated graph. Due to the COVID-1 situation with the Univerisity being closed, we are not able to do so at the moment. We are happy to share the data with anyone if requested, or if a possibility, to be added to the supplementary data at a later timepoint. Although in our opinion, the presentation of the individual strain will not add additional value with respect to the interpretation of the results as presented.
In discussion the authors state that „A. fumigatus hast he potential to undergo in-host adaptation to lysozymes“ (line 334). On the the other hand, they state that they „did not observe (such) a trend towards AMP resistance“ in their experimental setting (line 385 ff). This is confusing for the reader and should be clarified togther with experimental data.
> We hypothesise that in-host adaptation to AMPs is possible, despite not finding specific evidence within our isolates. However, as we explained in the manuscript, this may be because the isolates used were exposed to only minimal amounts of these AMPs in-host, or that AMP resistance is unlikely to develop. If the latter holds true, our findings support the development of AMP based antifungal therapy, as our data indicate that the acquisition of resistance is either not likely or a much slower process than the acquisition of antifungal drug resistance. In vitro experimental evolution experiments, during which the fungus is exposed to AMPs over a longer period, could provide useful insights into this.
The authors state that the „environmental isolates possessed on average lower metabolic activity after incubation with 20 μM lysozyme when compared to either chronic, acute or CF strains“ (line 200) and explain that as follows: „The environmental isolates are unlikely to have been exposed to lysozyme previously and this might account for their higher susceptibility to this AMP. This might indicate that A. fumigatus has the potential to undergo in-host adaptation to lysozyme. (line 332 ff). If so, should then A. fumigatus strains isolated from the patients with acute invasive aspergillosis show similar susceptibility pattern as the evironmental strains since it is most likely that the origin of these infections are enviromental strains and not „clinical“ strains? This should be clarified in more detail.
> The reviewer does raise a very interesting point, and we have therefore extended our discussion on this matter in lines 344-348:
‘It is worth noting that in theory, isolates from patients with acute invasive aspergillosis are likely to have spent a shorter time in-host compared to those from patients with chronic aspergillosis, and as such expected to show comparable susceptibilities to lysozyme as shown for the environmental isolates. However, we were not able to show this in our study, which is most likely explained by the complexity of underlying mechanisms involved in in-host adaptation.’
Why does the percentage of hyphae differ between the initial experiments (15 %, fig 5) and further tests (40%, Fig 6)?
> The initial characterisation experiment was performed separately from the further investigations. Hyphal growth is highly variable and slight changes in growth conditions (e.g. batches of media) can have a large impact on the growth velocity.
The authors refer to data from other groups showing that CF patients have higher levels of lysozyme and lactoferrin in bronchoalveolar lavage fluid in comparison to healthy controls. In addition, increased levels of β-defensin-1, β-defensin-2 and LL-37 have been described in newborns during pulmonary or systemic infection in comparison to healthy controls. Is there any data on the AMP levels in immunocompromised patients such as hematopoietic stem cell transplant recipients? What does that mean for the therapeutic use of AMPs against fungal infections? Which patient population could benefit from AMP anti-fungal therapy?
> We are not aware of any such literature, and unfortunately the questions asked can’t be answered with the results of the study presented. With regards to the therapeutic use of AMPs, this could be of benefit to all patients suffering from pulmonary aspergillosis, but if specific data would become available of particular patient groups with a proven lack or decrease of specific AMPs in the lung, more targeted treatment with AMPs would be an option.
Can the authors provide some references showing a correlation of AMP levels and severity/cource of infections?
> Unfortunately, we are not aware of any such literature showing this correlation.